# Improving Neural Network Training in Low Dimensional Random Bases

**Frithjof Gressmann**
Graphcore Research
Bristol, UK
frithjof@graphcore.ai

**Zach Eaton-Rosen**
Graphcore Research
London, UK
zacher@graphcore.ai

**Carlo Luschi**
Graphcore Research
Bristol, UK
carlo@graphcore.ai

## Abstract

Stochastic Gradient Descent (SGD) has proven to be remarkably effective in optimizing deep neural networks that employ ever-larger numbers of parameters. Yet, improving the efficiency of large-scale optimization remains a vital and highly active area of research. Recent work has shown that deep neural networks can be optimized in randomly-projected subspaces of much smaller dimensionality than their native parameter space. While such training is promising for more efficient and scalable optimization schemes, its practical application is limited by inferior optimization performance.

Here, we improve on recent random subspace approaches as follows: Firstly, we show that keeping the random projection fixed throughout training is detrimental to optimization. We propose re-drawing the random subspace at each step, which yields significantly better performance. We realize further improvements by applying independent projections to different parts of the network, making the approximation more efficient as network dimensionality grows. To implement these experiments, we leverage hardware-accelerated pseudo-random number generation to construct the random projections on-demand at every optimization step, allowing us to distribute the computation of independent random directions across multiple workers with shared random seeds. This yields significant reductions in memory and is up to $10\times$ faster for the workloads in question.

## 1 Introduction

Despite significant growth in the number of parameters used in deep learning networks, Stochastic Gradient Descent (SGD) continues to be remarkably effective at finding minima of the highly over-parameterized weight space [9]. However, empirical evidence suggests that not all of the gradient directions are required to sustain effective optimization and that the descent may happen in much smaller subspaces [14]. Many methods are able to greatly reduce model redundancy while achieving high task performance at a lower computational cost [34, 29, 5, 36, 17, 15, 32, 21].

Notably, Li et al. [26] proposed a simple approach to both quantify and drastically reduce parameter redundancy by constraining the optimization to a (fixed) randomly-projected subspace of much smaller dimensionality than the native parameter space. While the work demonstrated successful low-dimensional optimization, its inferior performance compared with standard SGD limits its practical application.

Here, we revisit optimization in low-dimensional random subspaces with the aim of improving its practical optimization performance. We show that while random subspace projections have computational benefits such as easy distribution on many workers, they become less efficient with growing projection dimensionality, or if the subspace projection is fixed throughout training. We observe that applying smaller independent random projections to different parts of the network and re-drawing them at every step significantly improves the obtained accuracy on fully-connected and several convolutional architectures, including ResNets on the MNIST, Fashion-MNIST and CIFAR-10 datasets.[1]

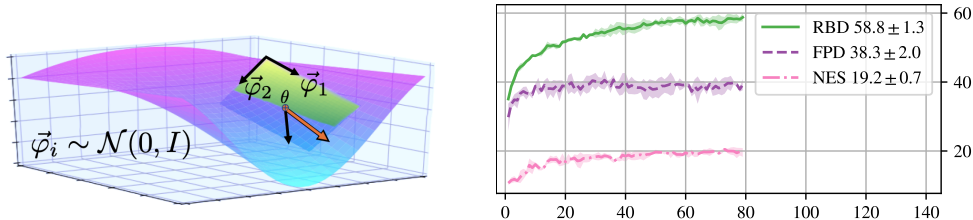

Figure 1: Left: schematic illustration of random subspace optimization on a 3D loss landscape. At the point $\theta$, the black arrow represents the direction of steepest descent computed by conventional SGD. The colored arrow represents the direction of steepest descent under the constraint of being in the chosen lower dimensional random subspace (the green plane). Right: Validation accuracy (y) against epochs (x) for CIFAR-10 classification of a $D = 78,330$ parameter ResNet-8 using low-dimensional optimization with $d = 500$. Our method RBD improves upon the same-$d$ FPD baseline [26] as well as black-box NES optimization [33] by 20.5% and 39.6% respectively.

## 2 Background

Using random projections to reduce dimensionality is appealing both computationally (fewer dimensions to operate on) and theoretically (due to their well-known approximation capabilities [20, 31, 13]). Sampled projections can map from sufficiently high-dimensional into low-dimensional Euclidean spaces without affecting the relative distances in a point sample. Thus, they can make investigations of certain complex structures more tractable [8]. Furthermore, they are computationally cheap to generate and can be easily stored, communicated, and re-created using random seeds [33].

### 2.1 Random projections for neural network optimization

Li et al. [26] utilized random projections to reduce the dimensionality of neural networks, aiming to quantify the difficulty of different tasks. Specifically, they constrained the network optimization into a fixed low-dimensional, randomly oriented hyper-plane to investigate how many dimensions are needed to reach 90% accuracy against the SGD baseline on a given task (Fixed Projection Descent, FPD). They found that this "intrinsic dimensionality" tended to be much lower than the parameter dimensionality of the network, suggesting that the network's solution space of the problem was smaller than its parameterization implied. Since FPD performs the parameter update in the fixed low-dimensional subspace (not native parameter space), the amount of trainable parameters is reduced significantly. However, the optimization progress is only defined with respect to the particular projection matrix and constrained to the subspace sampled at initialization.

### 2.2 Random projections for derivative-free gradient approximation

As opposed to using the analytical gradient of the loss with respect to sampled subspaces, random directions also allow for stochastic gradient-free estimation [30, 23, 4, 37]. Notably, variants of the derivative-free Natural Evolution Strategies (NES) have recently been applied to train large-scale image classification and reinforcement learning policy networks with hundreds of thousands of dimensions, suggesting that low-dimensional random directions may be rich enough to capture gradient information of high-dimensional neural networks [33, 25]. We discuss the formal connection of our approach with ES in Section A of the Supplementary Material.

## 2.3 Random projections for analysis of the loss landscape

Another application area of dimensionality reduction through random projection is the analysis of the neural network loss landscape. In its simplest form, the loss surface can be projected onto linear subspace by evaluating a number of loss values in random directions. Li et al. [27] conducted a comprehensive visualization of the loss landscapes for various architecture choices. They observed a transition from smooth to highly chaotic surfaces with increasing network depth. He et al. [18] analyzed the low-loss regions in "asymmetric" random directions to argue that the location of a solution within a valley correlates with its generalization ability. Random projection can also make computationally prohibitive forms of analysis more feasible. For instance, Fort and Scherlis [11] computed and analyzed the Hessian of lower-dimensional random hyperplanes of neural networks to investigate the relation of local curvature and the effectiveness of a given initialization. In follow-up work, Fort and Ganguli [10] proposed a model of a neural network's local geometry. As we will discuss in Section 4.5, our method allows for visualizations of the loss surface in the sampled random directions at every step to analyze the descent behavior.

## 2.4 Scalability and parallelization of random projections

Network training often leverages parallel computation of stochastic gradients on different workers that are exchanged and averaged at every step (*data parallel SGD*). However, inter-worker communication to exchange the locally-computed $D$-dimensional gradients can become a prohibitive bottleneck for practical distributed implementations [2, 36]. Notably, optimization in random subspaces can make the parallel execution on many workers more efficient, since random vectors are computationally cheap to generate and can be shared by the workers before the start of the training as a sequence of random seeds [33]. The locally-computed gradient can thus be communicated using only the low-dimensional gradient vector, while the high-dimensional update can be regenerated using the random projection. Our approach actively leverages this property to enable an efficient implementation of large random projections.

# 3 Method

In this section, we formally propose our method for low-dimensional optimization in random bases. We consider the following setting: Given a data distribution $\mathcal{Z} = \mathcal{X} \times \mathcal{Y}$ from which samples $(x, y)$ are drawn, we seek to learn the $D$ parameters $\boldsymbol{\theta} \in \mathbb{R}^D$ of a neural network $f : \mathbb{R}^{|X|} \times \mathbb{R}^D \to \mathbb{R}^{|Y|}$ by minimizing the empirical risk of the form $l(\boldsymbol{\theta}) = (1/M) \sum_{(x_i, y_i) \sim \mathcal{D}} L(f(x_i, \boldsymbol{\theta}), y_i)$, where $L$ denotes the network's loss on the samples $(x_i, y_i)$ of the training set $\mathcal{D}$ of size $M$.

SGD optimizes the model using a stochastic gradient $\boldsymbol{g}_t^{SGD} = \nabla_{\boldsymbol{\theta}} L(f(x_j, \boldsymbol{\theta}_t)), y_j)$ where $(x_j, y_j)$ are randomly drawn samples of $\mathcal{D}$ at timestep $t$. The weights $\boldsymbol{\theta}_t$ are adjusted iteratively following the update equation $\boldsymbol{\theta}_{t+1} := \boldsymbol{\theta}_t - \eta^{SGD} \boldsymbol{g}_t^{SGD}$ with learning rate $\eta^{SGD} > 0$. In the commonly used mini-batch version of SGD, the update gradient is formed as an average of gradients $\boldsymbol{g}_{t,B}^{SGD} = (1/B) \sum_{b=1}^{B} \nabla_{\boldsymbol{\theta}} L(f(x_b, \boldsymbol{\theta}_t), y_b))$ over a mini-batch sample of size $B$. To simplify the discussion, we have omitted the explicit notation of the mini-batch in the following gradient expressions.

## 3.1 Random bases descent

To reduce the network training dimensionality, we seek to project into a lower dimensional random subspace by applying a $D \times d$ random projection matrix $\mathbf{P}$ to the parameters $\boldsymbol{\theta}_t := \boldsymbol{\theta}_0 + \mathbf{P} \boldsymbol{c}_t$, where $\boldsymbol{\theta}_0$ denotes the network's initialization and $\boldsymbol{c}_t$ the low-dimensional trainable parameter vector of size $d$ with $d < D$. If $\mathbf{P}$'s column vectors are orthogonal and normalized, they form a randomly oriented base and $\boldsymbol{c}_t$ can be interpreted as coordinates in the subspace. As such, the construction can be used to train in a $d$-dimensional subspace of the network's original $D$-dimensional parameter space.

In this formulation, however, any optimization progress is constrained to the particular subspace that is determined by the network initialization and the projection matrix. To obtain a more general expression of subspace optimization, the random projection can instead be formulated as a constraint of the gradient descent in the original weight space. The constraint requires the gradient to be expressed in the random base $\boldsymbol{g}_t^{RB} := \sum_{i=1}^{d} c_{i,t} \boldsymbol{\varphi}_{i,t}$ with random basis vectors $\{\boldsymbol{\varphi}_{i,t} \in \mathbb{R}^D\}_{i=1}^{d}$

| **Random Bases Descent (RBD) Algorithm** | **Parallelized RBD** |
|---|---|
| **Input:** Learning rate $\eta^{RBD}$, network initialization $\boldsymbol{\theta}_{t=0}$ | **Input:** Learning rate $\eta^{RBD}$, network initialization $\boldsymbol{\theta}_{t=0}$ |
| | **Initialize**: $K$ workers with known random seeds |
| **for** $t = 0, 1, 2, \ldots$ | **for** $t = 0, 1, 2, \ldots$ |
| |   **for** each worker $k = 1, \ldots, K$ |
|   Sample random base $\{\boldsymbol{\varphi}_{1,t}, ..., \boldsymbol{\varphi}_{d,t}\}$, $\boldsymbol{\varphi}_{i,t} \in \mathbb{R}^D$ |     Sample random base $\{\boldsymbol{\varphi}_{1,t}^k, ..., \boldsymbol{\varphi}_{d,t}^k\}$, $\boldsymbol{\varphi}_{i,t}^k \in \mathbb{R}^D$ |
|   Reset coordinates $\boldsymbol{c}_t$ to $\vec{0}$ |     Reset coordinates $\boldsymbol{c}_t^k$ to $\vec{0}$ |
|   Compute coordinates $\boldsymbol{c}_{t+1} = \nabla_{\boldsymbol{c}} L(\boldsymbol{\theta}_t + \sum_i c_{i,t} \boldsymbol{\varphi}_{i,t})$ |     Compute coordinates $\boldsymbol{c}_{t+1}^k = \nabla_{\boldsymbol{c}} L(\boldsymbol{\theta}_t + \sum_i c_{i,t} \boldsymbol{\varphi}_{i,t}^k)$ |
| |   **end for** |
| |   Send all coordinates $\boldsymbol{c}_{t+1}^k$ from each worker to every other worker |
| |   **for** each worker $k = 1, \ldots, K$ |
| |     Reconstruct random bases $\boldsymbol{\varphi}_i^j$ for $j = 1, ..., K$ |
|   Update weights $\boldsymbol{\theta}_{t+1} = \boldsymbol{\theta}_t - \eta^{RBD} \sum_i c_{i,t+1} \boldsymbol{\varphi}_{i,t}$ |     Update weights $\boldsymbol{\theta}_{t+1} = \boldsymbol{\theta}_t - \eta^{RBD} \sum_{i,j} c_{i,t+1}^j \boldsymbol{\varphi}_{i,t}^j$ |
| |   **end for** |
| **end for** | **end for** |

**Algorithm 1:** Training procedures for a single worker (left) and for parallelized workers (right). Notably, the distributed implementation does not require a central main worker as the PRNG generation can be shared between workers in a decentralized way. As such, the algorithm entails a trade-off between increased compute through PRNG versus reduced communication between workers.

and coordinates $c_{i,t} \in \mathbb{R}$. The gradient step $\boldsymbol{g}_t^{RB} \in \mathbb{R}^D$ can be directly used for descent in the native weight space following the standard update equation $\boldsymbol{\theta}_{t+1} := \boldsymbol{\theta}_t - \eta_{RB}\, \boldsymbol{g}_t^{RB}$.

To obtain the $d$-dimensional coordinate vector, we redefine the training objective itself to implement the random bases constraint: $L^{RBD}(c_1, ..., c_d) := L(\boldsymbol{\theta}_t + \sum_{i=1}^d c_i \boldsymbol{\varphi}_{i,t})$. Computing the gradient of this modified objective with respect to $\boldsymbol{c} = [c_1, ..., c_d]^T$ at $\boldsymbol{c} = \vec{0}$ and substituting it back into the basis yields a descent gradient that is restricted to the specified set of basis vectors:
$\boldsymbol{g}_t^{RBD} := \sum_{i=1}^d \frac{\partial L^{RBD}}{\partial c_i}\Big|_{\boldsymbol{c}=\vec{0}} \boldsymbol{\varphi}_{i,t}$. The resulting optimization method is given in Algorithm 1. Note that this scheme never explicitly calculates a gradient with respect to $\boldsymbol{\theta}$, but performs the weight update using only the $d$ coordinate gradients in the respective base.

### 3.1.1 Compartmentalized approximation

The gradient expression allows not only for the adjustment of the basis vectors $\boldsymbol{\varphi}$ at different steps (time axis), but can also be partitioned across the $D$-dimensional parameter space (network axis).

Such partitioning of the network into different *compartments* can improve the approximation capability of the subspace. Consider approximating a $D$-dimensional vector $\boldsymbol{\alpha}$ in a random base $\{\boldsymbol{\varphi_i} \in \mathbb{R}^D\}_{i=1}^N$ with $N$ bounded coefficients: $\boldsymbol{\alpha} \approx \hat{\boldsymbol{\alpha}} = \sum_{i=1}^N c_i \boldsymbol{\varphi}_i$. As the dimensionality of $D$ grows, it may be required to generate exponentially many samples as the independently chosen random basis vectors become almost orthogonal with high probability [8]. For high-dimensional network architectures, it can thus be useful to reduce the *approximation dimensionality* through compartmentalization of the approximation target (the network parameters). For instance, $\boldsymbol{\alpha}$ could be partitioned into $K$ evenly sized compartments of dimension $Q = D/K$. The compartments $\{\boldsymbol{\alpha}_\kappa \in \mathbb{R}^Q\}_{\kappa=1}^K$ would be approximated with independent bases $\{\boldsymbol{\varphi}_{i,\kappa} \in \mathbb{R}^Q\}_{i=1}^{N/K}$ of reduced dimensionality: $\boldsymbol{\alpha}_\kappa \approx \hat{\boldsymbol{\alpha}}_\kappa = \sum_{i=1}^{N/K} c_{i,\kappa} \boldsymbol{\varphi}_{i,\kappa}$. While the overall number of coefficients and bases vectors remains unchanged, partitioning the space constrains the dimensionality of randomization, which can make random approximation much more efficient [13]. It can be instructive to consider extreme cases of compartmentalized approximations. If the $K \equiv D$, we recover the SGD gradient as every weight forms its own compartment with a trainable coefficient. If $d \equiv D$ and is large enough to form an orthogonal base, the optimization becomes a randomly rotated version of SGD.

Apart from evenly dividing the parameters, we construct compartmentalization schemes more closely related to the network architecture. For example, we use "layer-wise compartmentalization", where the parameter vector of each *layer* uses independent bases. The bases dimension in each compartment can also be adjusted dynamically based on the number of parameters in the compartment. If the number and dimension of the compartments are chosen appropriately, the total amount of trainable coefficients can be kept low compared to the network dimensionality.

# 4   Results and discussion

## 4.1   Experimental setup

The architectures we use in our experiments are: a fully-connected (FC) network, a convolutional neural network (CNN), and a residual network (ResNet) [19]. Since we have to generate random vectors of the network size in every training step, for initial evaluation, we choose architectures with a moderate ($\approx 10^5$) number of parameters. All networks use ReLU nonlinearities and are trained with a softmax cross-entropy loss on the image classification tasks MNIST, Fashion-MNIST (FMNIST), and CIFAR-10. Unless otherwise noted, basis vectors are drawn from a normal distribution and normalized. We do not explicitly orthogonalize, but rely on the quasi-orthogonality of random directions in high dimensions [13]. Further details can be found in the Supplementary Material.

## 4.2   Efficient random bases generation

As apparent from Algorithm 1, the implementation depends on an efficient generation of the $D \times d$ dimensional random subspace projections that are sampled at each step of the training. To meet the algorithmic demand for fast pseudo-random number generation (PRNG), we conduct these experiments using Graphcore's first generation Intelligence Processing Unit (IPU)[2]. The Colossus™ MK1 IPU (GC2) accelerator is composed of 1216 independent cores with in-core PRNG hardware units that can generate up to 944 billion random samples per second [22]. This allows us to generate the necessary random base vectors locally on the processor where the computation happens — substituting fast local compute for expensive communication. The full projection is never kept in memory, but encoded as a sequence of random seeds; samples are generated on demand during the forward-pass and re-generated from the same seed during the backward-pass. Although our TensorFlow implementation did not rely on any hardware-specific optimizations, the use of the hardware-accelerated PRNG proved to be essential for fast experimentation. On a single IPU, random bases descent training of the CIFAR-10 CNN with the extremely sample intensive dimension $d = 10$k achieved a throughput of 31 images per second (100 epochs / 1.88 days), whereas training the same model on an 80 core CPU machine achieved 2.6 images/second (100 epochs / 22.5 days). To rule out the possibility that the measured speedup can be attributed to the forward-backward acceleration only, we also measured the throughput of our implementation on a GPU V100 accelerator but found no significant throughput improvement relative to the CPU baseline.

## 4.3   Distributed implementation

The training can be accelerated further through parallelization on many workers. Like with SGD, training can be made data parallel, where workers compute gradients on different mini-batches and then average them. RBD can also be parallelized by having different workers compute gradients in different random bases. Gradients can be exchanged using less communication bandwidth than SGD, by communicating the low-dimensional coefficients and the random seed of the base. Notably, the parallelization scheme is not limited to RBD but can also be applied to the fixed random projection methods of [26]. In Figure 5, we investigate whether training performance is affected when distributing RBD. We observe constant training performance and almost linear wall-clock scaling for 16 workers.

## 4.4   Performance on baseline tasks

We begin by establishing performance on the fully-connected and convolutional architectures for $d = 250$ dimensions, which is the lowest reported intrinsic dimensionality for the image classification problems investigated by Li et al. [26]. Table 1 reports the validation accuracy after 100 epochs (plots in Supplementary Material, Figure B.6). All methods other than SGD use a dimensionality reduction factor of $400\times$. As expected, SGD achieves the best validation accuracy. The NES method displays high training variance while failing to achieve even modest performance in any task. FPD falls short of the SGD target by $\approx 20 - 40$ percentage points, while exceeding the performance of NES. By changing the basis at each step, RBD manages to decrease the FPD-SGD gap by up to 20%. Just like SGD, the random subspace training shows little variance between independent runs with different initializations.

Table 1: Validation accuracy after 100 epochs of random subspace training for dimensionality $d = 250$ compared with the unrestricted SGD baseline (mean $\pm$ standard deviation of 3 independent runs using data augmentation). To ease comparison with [26] who reported relative accuracies, we additionally denote the achieved accuracy as a fraction of the SGD baseline accuracy in parenthesis. Re-drawing the random subspace at every step (RBD) leads to better convergence than taking steps in a fixed randomly projected space of the same dimensionality (FPD). While training in the $400\times$ smaller subspace can almost match full-dimensional SGD on MNIST, it only reaches 78% of the SGD's baseline on the harder CIFAR-10 classification task. Black-box optimization using evolution strategies for the same dimensionality leads to far inferior optimization outcomes (NES). While NES's performance could be improved significantly with more samples (i.e. higher $d$), the discrepancy demonstrates an advantage of gradient-based subspace optimization in low dimensions.

| MODEL | NES | FPD | RBD | SGD |
|---|---|---|---|---|
| FC-MNIST | 22.5±1.7 (0.23) | 80±0.4 (0.81) | 93.893±0.024 (0.96) | 98.27±0.09 |
| FC-FMNIST | 45±6 (0.52) | 77.3±0.29 (0.89) | 85.65±0.2 (0.98) | 87.32±0.21 |
| FC-CIFAR10 | 17.8±0.5 (0.34) | 21.4±1.2 (0.41) | 43.77±0.22 (0.84) | 52.09±0.22 |
| CNN-MNIST | 51±6 (0.51) | 88.9±0.6 (0.89) | 97.17±0.1 (0.98) | 99.41±0.09 |
| CNN-FMNIST | 37±4 (0.4) | 77.8±1.6 (0.85) | 85.56±0.1 (0.93) | 91.95±0.18 |
| CNN-CIFAR10 | 20.3±1 (0.25) | 37.2±0.8 (0.46) | 54.64±0.33 (0.67) | 81.4±0.4 |

## 4.5 Properties of random bases optimization

It is worth noting that FPD tends to converge faster than other methods, but to a lower validation accuracy when compared to RBD. This suggests that fixing the basis is detrimental to learning. Having established baseline performances, we now move onto reducing the remaining performance gap to SGD.

**Influence of dimensionality** Intuitively, performance should improve when using a larger number of basis vectors as the subspace descent is able to better approximate the "true" gradient. To analyze this, we follow Zhang et al. [38] in quantifying the similarity of the RBD gradient and the unrestricted SGD gradient with the Pearson Sample Correlation Coefficient. For CIFAR-10 CNN training, we find that using more random basis directions improves the correlation of the RBD gradient with SGD, as well as the achieved final accuracy (see Figure B.7 in the Supplementary Material). However, a linear improvement in the gradient approximation requires an exponential increase in the number of subspace directions. As a result, achieving 90% of the SGD performance requires a subspace dimension $d = 10^4$, which represents almost a tenth of the original network dimension $D = 122$k.

From another perspective, the finding implies that random bases descent with very few directions remains remarkably competitive with higher dimensional approximation. In fact, we find that training is possible with as few as two directions, although convergence happens at a slower pace. This raises the question as to whether longer training can make up for fewer available directions at each step. To explore this question, we train the CNN on CIFAR-10 for 2000 epochs (2.5 million steps) where optimization with $d = 2$ random directions appears to converge and compare it with dimensionality $d = 10$ and $d = 50$ (full training curves are in Figure B.13 of the Supplementary Material). When trained to convergence, two random directions are sufficient to reach $48.20 \pm 0.23$% CIFAR-10 validation accuracy. However, training with 10 and 50 directions surpasses this limit, implying that considering multiple directions in each step provides an optimization advantage that cannot be compensated with more time steps (SGD converges to $68.06 \pm 0.14$% after training for the same number of epochs). The chosen dimensionality appears to determine the best possible accuracy at convergence, which is consistent with the intrinsic dimensionality findings of [26].

**Role of sampled bases** The effectiveness of the optimization not only relies on the dimensionality of the random subspace, but also on the type of directions that are explored and their utility for decreasing the loss. We experimented with using Gaussian, Uniform and Bernoulli generating functions for the random bases. We observed a clear ranking in performance: Gaussian $\gg$ Uniform $\gg$ Bernoulli, which held in every experiment (Table 2, plots in Figure B.15 in the Supplementary Material).

Table 2: Validation accuracy after 100 epochs training with different directional distributions, Uniform in range $[-1, 1]$, unit Gaussian, and zero-mean Bernoulli with probability $p = 0.5$ (denoted as Bernoulli-0.5). Compared to the Gaussian baseline, the optimization suffers under Uniform and Bernoulli distributions whose sampled directions concentrate in smaller fractions of the high-dimensional space.

| MODEL | BERNOULLI-0.5 | UNIFORM | NORMAL |
|---|---|---|---|
| FC-MNIST, D=101,770 | 77.5±0.4 | 89.61±0.29 | 94.95±0.11 |
| FC-FMNIST, D=101,770 | 76±0.6 | 81.7±0.14 | 85.26±0.2 |
| FC-CIFAR10, D=394,634 | 29.12±0.33 | 36.52±0.2 | 42.6±0.5 |
| CNN-MNIST, D=93,322 | 67±6 | 87±6 | 96.75±0.18 |
| CNN-FMNIST, D=93,322 | 65.9±3.3 | 77.8±1.9 | 83.8±0.7 |
| CNN-CIFAR10, D=122,570 | 28.5±0.6 | 41±1.4 | 52.3±0.9 |

To analyse this discrepancy, we visualize the local loss surface under the different distributions at different steps during optimization. As apparent in Figure 2, the local loss surface depends heavily on the choice of bases-generating function. In particular, we ascribe the superior performance of the Normal distribution to its ability to find loss-minimizing directions in the landscape. While the properties of the subspace loss surface are not guaranteed to carry over to the full-dimensional space [27], the analysis could inform an improved design of the subspace sampling and shed light on the directional properties that matter most for successful gradient descent.

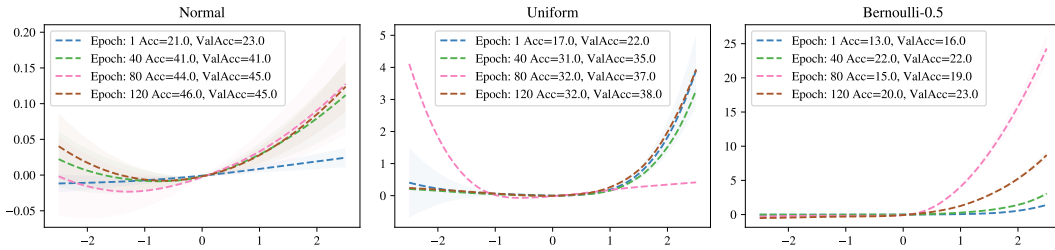

Figure 2: For different generating distributions, we plot the local loss landscape (y) at small displacements from the current weights (x) for an averaged sample of 25 directions at different epochs. Around zero (where the gradient is computed), the "flatness" of the local landscape in the random directions varies with the distribution. The Normal distribution has a pronounced slope at each epoch, meaning that the bases tend to represent useful descent directions. Uniform and Bernoulli each have flat local regions, hindering effective optimization.

**Relationship with SGD** An important question in the assessment of RBD is what type of descent behavior it produces in relation to full-dimensional SGD. While directly comparing the paths of optimizers in a high-dimensional space is challenging, we investigate potential differences in optimization behavior by testing whether we can "switch" between the optimizers without divergence. Specifically, we begin the optimization with RBD and switch to SGD at different points in training; vice versa, we start with SGD and switch to RBD. We do not tune the learning rates at the points where the optimizer switch occurs. We find that a switch between the low-dimensional RBD and standard SGD is possible at any point without divergence, and both update schemes are compatible at any point during training (Figure 4.5, further plots in Supplementary Material, Section B.5). After initial SGD training, RBD optimization continues to converge, but does not improve beyond the limit of training with just RBD. In fact, if the switch occurs at an accuracy level higher than the RBD accuracy limit, the RBD optimization regresses to RBD baseline level. Conversely, when training starts with RBD, SGD recovers its own baseline performance. The high compatibility of the low-dimensional and unrestricted descent suggests that the optimizers do not find disconnected regions of the loss landscape.

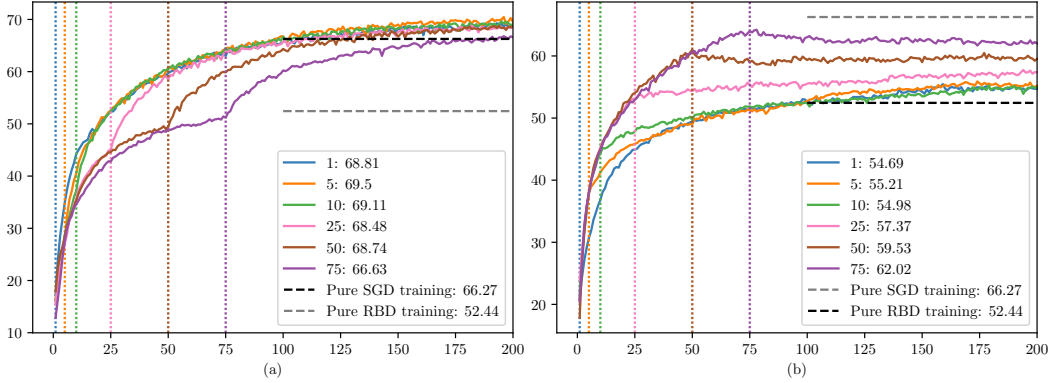

Figure 3: CIFAR10-CNN validation accuracy (y) against epochs (x) for hybrid training with switch after 1, 2, 5, 10, 25, 50, 75 epochs indicated by vertical rulers. (a) RBD followed by SGD and (b) SGD followed by RBD. Switching between the two optimizers is possible without divergence and yields the accuracy level of the single-optimizer baseline.

## 4.6 Improving approximation through compartmentalization

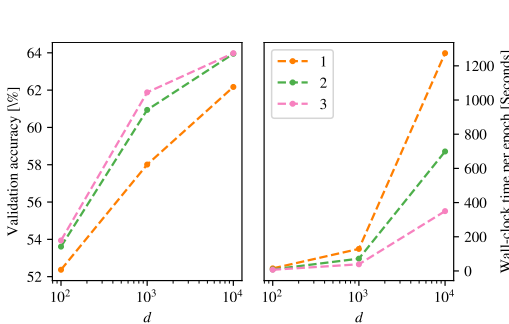

Figure 4: Investigating compartmentalization with CIFAR-10 CNN. Left: Validation accuracy under varying number of compartments and right: wall-clock time for these experiments. Compartmentalization both increases the accuracy and improves training time, most likely due to improved parallelization over compartments.

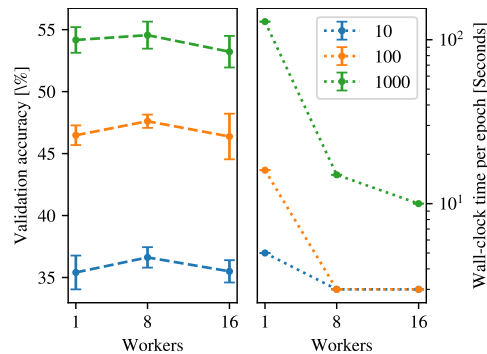

Figure 5: Investigating distributed RBD using CIFAR-10 CNN. Left: validation accuracy is constant whether using 1, 8 or 16 workers. Right: distributing the work leads to considerable reduction in wall-clock time for networks of varying dimensionality $d$.

As discussed in Section 3.1.1, constraining the dimensionality of randomization can make the approximation more efficient. To test this in our context, Figure 4 compares the performance of the network with differing numbers of compartments, with the network's parameters evenly split between compartments. We find benefits in both achieved accuracy and training time from this compartmentalization scheme. Notably, the scheme also boosts FPD performance, although it remains inferior to RBD (see Supplementary Material, Figure B.9).

To assess the viability of the random bases approach for deeper networks, we conduct experiments with ResNets [19]. To enable qualitative comparisons we use an 8-layered network with similar parameter count ($D = 78$k) at greater depth than the previously considered architectures. We compartmentalize the network layer-by-layer and allocate random bases coefficients proportionally to the number of weights in the layer. Table 3 shows the achieved accuracies after 80 epochs for varying reductions in the number of trainable parameters. RBD reaches 84% of the SGD baseline with a $10\times$ reduction of trainable parameters and outperforms FPD for all compression factors; even at $75\times$ reduction, its relative improvement is over 11%.

Table 3: Accuracies and correlation with full-dimensional SGD gradient for CIFAR-10 ResNet-8 training with data augmentation under varying numbers of trainable parameters.

| OPTIMIZER | $d$ | TRAINING ACC % | VALIDATION ACC % | CORRELATION |
|---|---|---|---|---|
| SGD | D=78330 | 88.89±0.13 | 83.84±0.34 | - |
| RBD, 10X REDUCTION | 7982 | 73.34±0.03 | 70.26± 0.02 | 0.42±0.18 |
| RBD, 25X REDUCTION | 3328 | 70.37±0.03 | 70.37± 0.01 | 0.40±0.20 |
| RBD, 50X REDUCTION | 1782 | 67.73±0.02 | 65.42± 0.02 | 0.38±0.22 |
| RDB, 75X REDUCTION | 1018 | 65.08±0.03 | 62.44± 0.04 | 0.38±0.23 |
| FPD, 10X REDUCTION | 7982 | 58.46±0.01 | 58.35±0.04 | 0.379±0.007 |

## 5  Conclusions and future work

In this work, we introduced an optimization scheme that restricts gradient descent to a few random directions, re-drawn at every step. This provides further evidence that viable solutions of neural network loss landscape can be found, even if only a small fraction of directions in the weight space are explored. In addition, we show that using compartmentalization to limit the dimensionality of the approximation can further improve task performance.

One limitation of the proposed method is that although our implementation can reduce communication significantly, generating the random matrices introduces additional cost. Future work could address this using more efficient sparse projection matrices [24, 28]. Furthermore, improved design of the directional sampling distribution could boost the descent efficiency and shed light on the directional properties that matter most for successful gradient descent. Many existing techniques that have been developed for randomized optimization such as covariance matrix adaptation [16] or explicit directional orthogonalization [7], among others, can be considered. Moreover, it is likely that non-linear subspace constructions could increase the expressiveness of the random approximation to enable more effective descent.

Given the ease of distribution, we see potential for training much larger models — where the size of the subspace is more akin to the size of today's "large models". The advent of hardware-accelerated PRNG may mean that randomized optimization strategies are well-poised to take advantage of massively parallel compute environments for the efficient optimization of very large-scale models.

### Broader Impact

The reduced communication costs of the proposed method may lead to more energy-efficient ways to distribute computation, which are needed to address growing environmental impacts of deep learning [35, 12]. For language modelling in particular, we still seem to be in the regime where bigger models perform better [6]. New optimization approaches could help reducing the energy consumption of training ever larger models. However, Jevon's Paradox suggests that increased efficiency may, counterintuitively, lead to increased total energy usage.

### Acknowledgments and Disclosure of Funding

There are no financial conflicts of interest to disclose. We thank Ivan Chelombiev, Anastasia Dietrich, Seth Nabarro, Badreddine Noune and the wider research team at Graphcore for insightful discussions and suggestions. We would also like to thank the anonymous reviewers for their contributions to this manuscript. Furthermore, we are grateful to Manuele Sigona and Graham Horn for providing technical support.

## Footnotes

[1]Our source code is available at https://github.com/graphcore-research/random-bases

[2]https://graphcore.ai

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
