[Supplementary Material]



# SUPPLEMENTARY MATERIAL

## A  Relationship with Evolution Strategies (ES)

In the main paper, we restrict the gradient to the random base

$$\boldsymbol{g}_t^{RB} := \sum_{i=1}^{d} c_{i,t}\, \boldsymbol{\varphi}_{i,t}\,.$$

Formally, this constraint also applies to special cases of Natural Evolution Strategies [37, 3]. The evolutionary random bases gradient can be derived from the Taylor expansion of the loss under a perturbation $\boldsymbol{\varphi}$, where $\mathrm{H}(\boldsymbol{\theta})$ denotes the Hessian matrix:

$$L(\boldsymbol{\theta} + \boldsymbol{\varphi})\boldsymbol{\varphi} = L(\boldsymbol{\theta})\boldsymbol{\varphi} + \boldsymbol{\varphi}^\top \nabla L(\boldsymbol{\theta})\boldsymbol{\varphi} + \frac{1}{2!}\boldsymbol{\varphi}^\top \mathrm{H}(\boldsymbol{\theta})\boldsymbol{\varphi}^2 + O(\boldsymbol{\varphi}^4)\,.$$

Taking the expectation with respect to perturbations drawn from a normal distribution with zero mean and constant variance $\sigma^2$ results in the odd central moments of the Gaussians vanishing and leaves the gradient estimator

$$\nabla L(\boldsymbol{\theta}_t) \approx \frac{1}{\sigma^2}\mathbb{E}_{\boldsymbol{\varphi}\sim\mathcal{N}(0,\mathbb{I})}[L(\boldsymbol{\theta}_t + \boldsymbol{\varphi})\boldsymbol{\varphi}]\,.$$

The sample approximation of the expected value then takes the form of a random basis with unit Gaussian base vectors $\boldsymbol{\varphi}_n \sim \mathcal{N}(0,\mathbb{I})$ and coordinates $c_n = L(\boldsymbol{\theta}_t + \sigma\boldsymbol{\varphi}_n)\sigma^{-1}d^{-1}$

$$\boldsymbol{g}_t^{ES} := \sum_{n=1}^{d} \frac{L(\boldsymbol{\theta}_t + \sigma\boldsymbol{\varphi}_n)}{\sigma\,d}\boldsymbol{\varphi}_n\,.$$

Similar estimators can be obtained for other symmetric distributions with finite second moment. A practical downside of $\boldsymbol{g}_t^{ES}$, however, is the high computational cost of the independent neural network loss evaluations that are required to obtain the coordinates $c_n$. Moreover, the additional hyper-parameter $\sigma$ that determines the magnitude of the perturbation needs to be carefully chosen [33]. Finally, it is worth noticing that the different evolutionary loss samples $c_n \propto L(\boldsymbol{\theta}_t + \sigma\boldsymbol{\varphi}_n)$ are typically evaluated on different mini-batches [38], whereas the approach discussed in this paper computes the coefficients on the same mini-batch.

## B  Further results

### B.1  Convergence behaviour of random bases training

Figure B.6 provides the validation curves of different the random subspace methods for the baseline dimensionality $d = 250$.

### B.2  Approximation with growing dimensionality

As discussed in Section 4.5, the performance of random bases descent improves when using a larger number of basis vectors. Figure B.7 quantifies the approximation quality in terms of achieved accuracy as well as correlation with the SGD gradient for an increasing number of base dimensions. A linear improvement in the gradient approximation requires an exponential increase in the number of subspace directions.

### B.3  Quasi-orthogonality of random bases

While increasing the number of random directions improves RBD's gradient approximation, the observed exponential growth of required samples makes reaching SGD-level performance difficult in

Figure B.6: Validation accuracy (y) against epochs (x) of random subspace training for dimensionality $d = 250$ compared with the unrestricted SGD baseline (average of 3 independent runs using data augmentation with report of standard deviation). Re-drawing the random subspace at every step (RBD) leads to better convergence than taking steps in a fixed randomly projected space of the same dimensionality (FPD). Black-box optimization using evolution strategies for the same dimensionality exhibits higher variance and leads to inferior optimization outcomes (NES).

Figure B.7: Validation accuracy after 100 epochs and mean gradient correlation with SGD plotted against increasing subspace dimensionality $d$ on the CIFAR-10 CNN (average of three runs). The gradient approximation quality and resulting accuracy only improves at a logarithmic rate and requires $d = 10^4$ subspace dimensions to achieve 90% of SGD accuracy.

practice. The diminishing returns of increasing the number of random bases may stem from the fact that high-dimensional random vectors become almost orthogonal with high probability; capturing relevant directions may thus become harder as approximation dimensionality grows [13].

We quantify the orthogonality of the base vectors with increasing dimensionality in Figure B.8. As expected, the mean cosine similarity across 100 pairs of random vectors decreases with growing dimensionality. For vectors with $\approx 10^5$ directions, as we use in these experiments, the mean cosine similarity is around $0.02$, and tends further towards zero for higher dimensions. This suggests that explicit orthogonalization techniques might improve the approximation capabilities of the random bases for smaller or compartmentalized networks (for further discussion see Choromanski et al. [7]).

## B.4 Compartmentalization

A simple way to limit the approximation dimensionality is compartmentalization, as discussed in Section 3.1.1. As shown in Figure 4 in the paper, compartmentalization can improve RBD's approximation ability as well as reduce wall-clock time requirements.

Figure B.8: Mean cosine similarity computed for 100 pairs of random vectors plotted for growing dimensionality. The error bars represent the standard deviation of the 100 cosine similarities. As the dimensionality increases, the vector's linear dependence decreases in line with the theoretical expectation [13]. The figure also implies that for the dimensionality that is considered in this work (order $10^5$), the random base vectors are not strictly orthogonal. It is thus possible that explicit orthogonalization could yield better approximation results.

### B.4.1 Compartmentalized FPD

We test the effect of compartmentalization when used in combination with FPD [26]. Figure B.9 shows that compartmentalization improves the achieved FPD accuracy, although the final accuracies are lower than for RBD. This suggests that compartmentalization provides optimization benefits that are independent of a timestep-dependent base.

Figure B.9: Investigating compartmentalization with CIFAR-10 CNN for Fixed Projection Descent [26]. Left: Validation accuracy under varying number of compartments and right: wall-clock time for these experiments. Like in the case of RBD, compartmentalization increases the achieved accuracy for a given dimensionality (albeit falling short of RBD accuracy level).

### B.4.2 Layer compartmentalization

Apart from splitting the network into evenly-sized parts, it is a natural idea to compartmentalize in a way that reflects the network architecture. We test this on the 5-layer CNN by splitting the random base at each layer into independent bases with 250 coordinates. For comparison, we train with uncompartmentalized base vectors using $5 \times 250 = 1250$ coordinates such that the overall amount of trainable parameters is preserved (see Figure B.10). We find that compartmentalizing the CNN network in such a way improves the achieved validation accuracy by $0.64 \pm 0.27$, $1.02 \pm 0.17$ and $3.85 \pm 0.64$ percent on MNIST, FMNIST and CIFAR-10 respectively.

Figure B.10: Validation accuracy (y) versus epochs (x) for CNN architectures with compartmentalized bases compared to the uncompartmentalized baseline. The random bases are compartmentalized per layer, i.e. each layer uses an independent random base with $d_\lambda = 250$ for the gradient approximation. The overall number of parameters $d = 5 \times 250 = 1250$ equals the number of trainable parameters of the uncompartmentalized baseline. Splitting the random base dimensionality across layers yields performance improvements which suggests that a reduced approximation dimension can be beneficial to optimization.

## B.5 Relationship with SGD

Figure B.11: Validation accuracy (y) versus epochs (x) for hybrid training where the first $q$ epochs use random bases descent while remaining updates are standard SGD (switch points $q$ are indicated by the vertical dashed rules of same color, $q \in \{1, 2, 5, 10, 25, 50, 75\}$). Switching between the optimizers is possible at any point and the SGD optimization in the later part of the training recovers the performance of pure SGD training (indicated by dark horizontal dashed line).

## B.6 Convergence with low dimensionality

Random bases descent with very few directions remains remarkably competitive with higher dimensional approximation. In fact, we find that training is possible with as few as two directions, although convergence happens at a slower pace. This raises the question as to whether longer training can make up for fewer available directions at each step. Such a trade-off would, for example, hold for a random walk on a convex bowl where the steps in random directions use a step size proportional to the slope encountered in the random direction. Lucky draws of directions that point to the bottom of the bowl will lead to a quick descent, while unlucky draws of flat directions will slow progress.

Figure B.12: Validation accuracy (y) versus epochs (x) for hybrid training where first q = 1, 2, 5, 10, 25, 50, 75 epochs use standard SGD while remaining epochs use random bases descent (switch points are indicated by the vertical dashed rules of the same color). Random bases descent (RBD) can sustain the optimization progress after the switch and converges to the accuracy level of pure RBD training (indicated by the dark horizontal dashed line). However, if the switch occurs at an accuracy level higher than the pure RBD training baseline, the optimization progress regresses to this lower accuracy.

In this setting, drawing more directions at every step and allowing more timesteps overall will both increase the likelihood of finding useful descent directions. However, in practice, we find that training on CIFAR-10 for longer does not make up for reduced dimensionality of the random base (see Figures B.13 B.14 for CNN and ResNet-8 respectively. The chosen dimensionality determines the achieved accuracy at convergence.

Figure B.13: CNN training on CIFAR-10 for 2000 epochs with low dimensionality. Each curve uses a tuned learning rate such that results are compared for the best respective validation loss. Two random directions are sufficient to reach an accuracy level of $48.20 \pm 0.23$ percent. However, the converged $d = 2$ training process is outperformed with $d = 10$ and $d = 50$. Overall, training for longer does not close the gap with the SGD baseline.

## B.7 Comparison of directional distributions

The effectiveness of the RBD's descent is highly dependent on the choice of directions. To illustrate this, consider the case where the first direction vector $\varphi_{i=0,t}$ is the actual full-dimensional SGD gradient; in this setting, SGD-like descent can be recovered through the use of Kronecker delta

Figure B.14: Random bases optimization (RBD) of ResNet-8 for 2000 epochs with low dimensionality. Training is possible with $d = 2$ directions but does not reach the accuracy level of 2-dimensional RBD on the simpler CNN architecture. However, if the random bases approximation is compartmentalized at each layer ($d = 42$), training reaches levels of accuracy that are comparable with the CNN baseline for a similar amount of trainable parameters $d = 50$. Training for longer does not make up for a reduced number of dimensions $d$.

coordinates $c_{i,t} = \delta_{i,t}$. We experimented with adjusting the generating distribution for the random bases using Gaussian, Uniform, and Bernoulli distributions. In high dimensions, unit Gaussians isotropically cover a sphere around the point in the parameter space, whereas directions drawn from a uniform distribution concentrate in directions pointing towards the corners of a hypercube (breaking rotational invariance). Likewise, directions drawn from a Bernoulli distribution are restricted by their discrete binary coordinates. On training with these different directional distributions, we observe a clear ranking in performance across used networks and datasets: Gaussian consistently outperforms Uniform directions, which itself outperform Bernoulli samples (see Figure B.15).

Figure B.15: Validation accuracy (y) for training over a 100 epochs (x) with different directional distributions UNIFORM in range $[-1, 1]$, unit GAUSSIAN, and zero-mean BERNOULLI with probability $p = 0.5$ (denoted as Bernoulli-0.5). Compared to the Gaussian baseline, the optimization suffers under Uniform and Bernoulli distributions whose sampled directions concentrate in smaller fractions of the high-dimensional space.

## C  Implementation details

### C.1  Details on the datasets and networks

We use the (Fashion-)MNIST and CIFAR-10 dataset as provided by the TensorFlow Datasets API with inputs 28x28x1 and 32x32x3 respectively [1]. The CIFAR images are normalized to standard mean and variance using tf.image.per_image_standardization. We apply the following data augmentation in each mini-batch. (F)MNIST: Pad 6 pixels on each dimension of the feature map and apply random cropping back to original dimensions followed by random brightness adjustments (tf.image.random_brightness). CIFAR-10: Add 4 pixels padding and apply random cropping back to the original dimension, followed by random left-right flips (tf.image.random_flip_left_right). The datasets are used with the following networks:

- **Fully-connected (FC)** with one hidden layer of width 128 resulting in a total number of parameters of $D = 101,770$ and $D = 394,634$ for (F)MNIST and CIFAR-10 respectively.
- **Convolutional (CNN)** with the following hidden layers: Conv (3x3, valid) 32 outputs - max pooling (2x2) - Conv (3x3, valid) 64 outputs - max pooling (2x2) - Conv (3x3, valid) 64 outputs - 64 fully connected. This results in an (F)MNIST-dimension of $D = 93,322$ and a CIFAR-10-dimension of $D = 122,570$.
- **ResNets** We use a ResNet [19] standard implementation provided by the Keras project at https://github.com/keras-team/keras-applications. The CIFAR-10 version of ResNet-8 and ResNet-32 have a dimensionality of $D = 78,330$ and $D = 467,194$ respectively.

Learning rates are tuned over powers of 2 in the range $[7, -19]$ for each particular combination of network and dataset. Tuning uses the training data split only, i.e. training on a 75% random split of the training data and selecting for the lowest loss on the 25% held-out samples. All experiments use a batch size of 32.

## D  List of hyperparameters

We present a list of all relevant hyperparameters. More details can be found in the source code that has been released at https://github.com/graphcore-research/random-bases.

All experiments use a batch size of 32. We do not use momentum or learning rate schedules. Table 4 lists the standard learning rates for a basis dimension of $d = 250$. The learning rate can be scaled down as the dimensionality increases, which suggests that the variance of the gradient approximation decreases. For instance, the dimensionality scaling in Figure B.7 used the power-2 learning rates -1, -2, -3, -5 for the dimensions $d = 10, 100, 1000, 10000$ respectively. Learning rates have to be adjusted when different distributions are used; for example, the Normal, Bernoulli and Uniform distribution in Figure B.15 use the learning rates $2^{-3}$, $2^{-1}$ and 2 respectively.

Table 4: Learning rates of proposed random bases descent and fixed projection descent baseline by [26] for the dimensionality $d = 250$, as well as standard SGD learning rates. All learning rates are denoted as powers of 2 (i.e. $-1 \to 2^{-1} = 0.5$).

| NETWORK | DATA SET | RANDOM BASES DESCENT | FIXED PROJECTION | SGD |
|---|---|---|---|---|
| FC | MNIST | 1 | -1 | -8 |
| | FMNIST | -1 | -1 | -7 |
| | CIFAR-10 | -5 | -5 | -12 |
| CNN | MNIST | -1 | -3 | -10 |
| | FMNIST | -3 | -1 | -9 |
| | CIFAR-10 | -3 | -1 | -11 |
| RESNET-8, $d = 250$ | CIFAR-10 | 3 | - | -3 |