[Reviews · NeurIPS 2020]

Review 1

Summary and Contributions: The paper propose several improvements on training with random subspaces: (1) re-drawing the subspace at each step, (2) diverse project across the network. Leveraging hardware accelerated pseudo-RNG, they claim that these contributions lead to significant performance improvements over the baseline approach from Li et al. termed as FPD. [after rebuttal] Thank you for taking the time to clarify the baseline metrics compared to FPD, I know see the improvement. For this paper, I would strongly suggest using the same metric choice as [23], and also draw the data directly from [23], otherwise its difficult to see if the paper has faithfully reproduced the baselines. As the main weakness of my review has been satisfied, I have raised my score.

Strengths: Novel training methods are of substantial relevance to the machine learning community, and the proposed method is more easily parallelizable across workers. The contributions are well motivated, and novel relative to prior work. The finding of dependence of the loss landscape on the bases-generating function can inspire future work in the area.

Weaknesses: The paper would be improved by empirical evaluations that better reflect the prior art. For example, in Figure 2 the authors put the FC-MNIST baseline for FPD at 80% validation accuracy (D=100K), but Li et al show that their method greatly exceeds that value (Figure 4, Li et al paper). Without a direct comparison with accuracy obtained from the prior art, this reviewer finds it difficult to validate the empirical data on performance improvements. While novel, the significance of the contributions could be improved by a stronger analysis of whether this method, by virtue of its potentially decreased wall clock time, can allow for larger networks beyond the current limit.

Correctness: The method is correct, but as noticed above, unclear if the empirical methodology appropriately measures comparisons to prior art.

Clarity: The presentation could be improved by: - More legible graphs. The legends are extremely small. I would recommend turning Figure 2 into a table instead (and not showing the accuracy against epochs, as it does not add much value). - In Figure 4 (right), its difficult to extract the scaling efficiency due to the times for 8-16 workers being squashed. In addition, is there any comparison to FPD approach in terms of parallelization? - Some of the interesting results are buried in the main text itself, and not as a table. For example, the validation accuracies of different generating functions in line 217. Most of the results are also in the Supplementary. The background could be less verbose, and the more interesting Supplementary results included as tables. - While interesting, the "Applications" header is confusingly organized. For more clarity, I would embed 4.7 in the main analysis when describing the generating function results rather than splitting them separately. I would also consolidate the two sections on HW-related acceleration (4.2 and 4.6) into one section describing how this model interacts with the hardware and systems. - The abstract could be improved by including more specific claims on the final optimization performance achieved and how it compares to prior work.

Relation to Prior Work: Yes, the contributions are differentiated against prior work, particularly Li et al.

Reproducibility: Yes

Additional Feedback:


Review 2

Summary and Contributions: In this manuscript, authors discuss how to improve the subspace projection based optimization methods for deep neural networks. The random subspace is re-generated at each iteration, instead of keeping constant in the previous work, which improves the performance significantly. Network parameters are further divided into multiple groups and projected independently to make the approximation more efficient. Hardware-accelerated pseudo-random number generation is adopted to improve the computational efficiency for the on-the-fly random projection.

Strengths: 1. The proposed method can be viewed as the further improvement for the previous work by Li et al. [23]. The modification is simple yet effective, which is to explore more random subspace during the training process, instead of keeping it constant. The update rule for model parameters is re-designed to allow for changes in the random subspace. The generation of pseudo-random numbers is implemented using Graphcore’s IPU hardware, which provides in-core PRNG units. This greatly improves the efficiency for generating random subspace at each training step. The resulting projection does not need to be stored in the memory; instead, it can be re-generated at both forward and backward passes.

Weaknesses: 1.Section 4.2. Authors mentioned that on a single IPU, the proposed method achieves the throughput of 31 images per second, while on a 80-core CPU machine, the throughput is 2.6 images per second. Which contributes more to this speed, the forward-backward passes or the pseudo-random number generation? The IPU hardware should be able to speed-up the forward & backward pass computation, compared against the standard CPU hardware, right? 2.Section 4.2. The throughput (31 images per second) seems do not match with the “100 epochs / 67 minutes” statement. The CIFAR-10 dataset consists of 50k training images, and if the validation time is omitted, then 100 epochs should take 50k * 100 / (31 * 60) = 2688 minutes. Did I miss something? 3.Scalability for more difficult prediction tasks. The previous work [23] tested their method on more tasks, including ImageNet classification and some other RL tasks. The ImageNet classification results indicate that the intrinsic dimension is quite large (>500k). Since the proposed method have improved [23] on MNIST & CIFAR-10, is it possible to extend it to ImageNet, or is it still too challenging for random subspace based optimization methods? Section 4.4. For comparison, Gaussian, uniform, and Bernoulli distributions are used to generated pseudo-random numbers. The last one, Bernoulli distribution, does it have some connections with those gradient sparsification methods (to reduce the communication overhead)?

Correctness: Yes

Clarity: Yes

Relation to Prior Work: Yes

Reproducibility: Yes

Additional Feedback: Please address issues listed in the “Weaknesses” section. ---- Authors have clarified some writing issues and provided additional throughput comparison results on PRNG & forward-backward computation on IPU/CPU/GPU. It remains unclear whether the proposed methods can be applied to more challenging tasks (e.g. ImageNet classification). However, it has already provided significant improvement over the previous FPD method, and should be able to inspire more future works in this area. I would like to raise my score to 6.


Review 3

Summary and Contributions: This paper pursues to answer similar questions the Fixed Projection Descent (FPD) [23] and shows that better qualitative results can be achieved with optimising across continually resampled random bases instead of a fixed single random projection, as the FPD does. This effectively means that the random projection is different at each optimisations step and as such it can lead to better exploration of the optimisation landscape. In a way this is not a surprising result, as the model dimensionality does not change, but it shines new light on the question of what the dimensionality of the gradient update is. Due to the fact that the gradient update subspace can be simply represented by a random seed it has interesting properties with regards to parallelisation.

Strengths: This paper is really well written and every decision is well motivated with plethora of interesting existing literature. The relation to ES, provided in the supplementary material is intriguing and shows that this method is somewhat bridging SGD and ES. It is really interesting that the random base optimisation is so well correlated with the SGD gradients, which almost never the case for other optimisation methods based on random sampling, especially those based on black-box optimisation. Thus this method might become a useful tool for analysing properties of SGD optimisation.

Weaknesses: What I found most worrying about this paper is that the FPD CIFAR-10 results does not seem to be consistent with the FPD paper [23]. In [23] the FPD appears to be able to achieve 90% of the original performance with 20 fold reduction of the parameters for the LeNet model (Table 1 in [23]), while Table 1 of this manuscript gets only 60% of performance with only 10 fold reduction of parameters. Similarly, [23] mentions that the ResNet appears to be more parameter efficient than the LeNet architecture, which indicates that FPD should generally work much better in this case. This makes me wonder if there is some underlying issue in the author's implementation? If so, it might be possible that if the FPD baseline is fixed, the observed improvement of the RBD method would not hold? Is this method going to provide a new insight to the intrinsic dimensionality of the gradient updates instead of the model itself? Of course, it is possible that there is something wrong with the evaluation in [23], however in that case it would be useful to address these discrepancies. Additionally, I was unable to find more details about the NES baseline parameters. I suppose for each update they used small number of random samples, thus the really low performance? The NES algorithm performance is extremely dependent on the number of samples per update, as shown in [30]. In fact, from information theory perspective, the number of samples used for each update in ES is also related to the dimensionality of the gradient, which is what this method is trying to show as well. Authors correctly point out that this method leads to more expensive computational cost due to increased amount of random numbers which need to be generated. However, it might be useful to show this in Figure 3 as well, for more clarity. This is also a potential bottleneck in the distributed version of the training, as all the random numbers need to be generated on the main worker, which might limit the potential usefulness of this approach towards reducing the computational cost. However, this is only minor issue which is probably a question of future research.

Correctness: It appears to me that the method and the empirical methodology is correct. However, I'm not certain how the results on the CIFAR-10 relate to results reported in other literature.

Clarity: This paper is really well written.

Relation to Prior Work: Yes, to the best of my knowledge this paper does provide discussion of relevant methods.

Reproducibility: Yes

Additional Feedback: It should probably be pointed out that this method does not allow compressing the number of parameters of a network, compared to the FBD (however, it's possible that I've missed that in the text). ___ Rebuttal ___ I would like to thank the authors for the rebuttal and for addressing my questions. I agree with their point that my main concerns with regards to comparison with [23] were a misunderstanding, and yes, the results seem to show a significant improvement in the performance. I agree with the authors that adding the relative improvement scores is going to help comparison between the works. As such, I am increasing my rating.

[Author Response · NeurIPS 2020]

We thank the reviewers for their time and productive comments. However, there is one misunderstanding that might have affected the reviewers' scores: reviewers 1 and 3 have both misunderstood the baseline. When compared like-for-like, our results outperform the baseline by a large margin (details below). While this misunderstanding is surely a shortcoming of our presentation, the reviewers criticised that we did not improve on baseline performance, when we in fact did. Leaving the misunderstanding aside, reviewers found the work to be "really well written and every decision is well motivated" (Reviewer #3), found that the proposed implementation "greatly improves the efficiency for generating random subspace at each training step" (Reviewer #2) and wrote that the contributions are "novel relative to prior work [...] and can inspire future work in the area" (Reviewer #1) without mentioning any other major concern. Given a correct understanding of the baseline, it seems likely that their overall scoring would have been more positive.

To expand further on the misunderstanding, Reviewers #1 and #3 pointed out that our results did not seem to be consistent with the results published by Li et al. [1]. The misunderstanding most likely stems from the fact that Li et al. [1] reported the achieved accuracy as a percent value relative to the SGD baseline while we reported the absolute percent accuracy, without normalising against the SGD baseline. In particular, for a 20x reduced CIFAR-10 LetNet, [1] reported 90% of the original 58% SGD accuracy which amounts to an $0.9 \cdot 57\% = 51.3\%$ accuracy in absolute terms (see Figure S14b in [1] which presents the absolute accuracies). This is consistent with our reported 58.35% accuracy for a 10x reduced Resnet-8-CIFAR-10 in Table 1, where the 7% improvement can be attributed to increased efficiency of the ResNet architecture, as Reviewer #3 expected. Similarly, our MNIST baseline of 80% reported in Figure 1 for d=250 is consistent with the absolute accuracy reported in Figure S6 in [1] (note that the network dimensionality is D=100K, but the subspace dimensionality is d=250 only). We realize now that our reporting should have made this subtle issue of the percent notation in [1] clearer. We will add the relative accuracy levels to the manuscript to ease the direct comparison with prior art.

*Minor*: All suggested improvements are gratefully received and we will incorporate the feedback into our revision.

* Reviewer #2 asked whether the substantial IPU hardware speedup over CPUs was due to the accelerated PRNG or could be merely explained by the forward-backward pass acceleration.

> While the IPU accelerates the forward-backward pass of the network, we found that the main bottleneck on CPU hardware is indeed the PRNG (particularly for large subspace dimensions d > 1000). To rule out the possibility that the measured speedup can be attributed to the forward-backward acceleration only, we benchmarked the throughput of our implementation on a GPU V100 accelerator that, unlike the IPU, does not have an on-chip PRNG. We found that the GPU provided no throughput improvement relative to the CPU baseline. We will include these additional results in the respective Section 4.2.

* Reviewer #2 noted that the throughput "31 images per second" does not match with the "100 epochs / 67 minutes" statement in Section 4.2.

> Thank you, this is a mistake. We accidentally mixed the images per second throughput for d=10k with the wall-clock time figure for d=1k. The correct throughput for d=1k is 1366 and 112 images per second on IPU and CPU respectively.

* Reviewer #1 asked if there was "any comparison to FPD approach in terms of parallelization?"

> Both approaches can be parallelized in the same way since FPD can be seen as a special case of our algorithm where $\varphi_t \equiv \varphi_0$. Our more efficient distributed implementation can thus be seen as a technical contribution that can also benefit the investigations of intrinsic dimensionality in [1]. We will update the discussion to point this out.

* Reviewer #3 asked whether the low performance of the NES baseline stems from a small number of random samples.

> Indeed, the NES baseline in Figure 2 used the same very low-dimensional number of d=250 samples, while more samples would certainly increase the approximation quality. The low-dimensional comparison at d=250 demonstrates the superiority of gradient based RBD optimization over NES black-box sampling in this setting. We will adjust the caption of Figure 2 to underline this point.

* Reviewer #3 asked about a "potential bottleneck in the distributed version of the training, as all the random numbers need to be generated on the main worker".

> This is an good observation that motivates a trade-off between increased compute through PRNG versus reduced communication between workers. Notably, however, our implementation does not require a central main worker but the PRNG generation can be shared between workers in a decentralised way to load balance potential PRNG bottlenecks (see Algorithm 1, right).

[1]    Chunyuan Li et al. "Measuring the Intrinsic Dimension of Objective Landscapes". In: Sixth International Conference on Learning Representations. 2018. URL: https://openreview.net/forum?id=ryup8-WCW.


[Meta-Review · NeurIPS 2020]

The reviewers were consistent in their appreciation of the paper, as the paper demonstrated clear improvements over the ICLR work [23], and the inconsistencies that originally worried some reviewers were clarified by the rebuttal. Drawing a new subspace every iteration appears to be novel for the neural net application, though the authors point out connections with ES (and the AC notes connections to DFO community literature as well, e.g., https://arxiv.org/abs/2003.02684 and https://arxiv.org/abs/1905.01332). The reviewers also liked the compartmentalization idea. To summarize, though the initial reviewers response was only mildly positive, after the rebuttal and our discussions, the reviewers think this paper empirically shows a significant improvement over prior work.